# Computational Probing the Methylation Sites Related to EGFR Inhibitor-Responsive Genes

**DOI:** 10.3390/biom11071042

**Published:** 2021-07-16

**Authors:** Rui Yuan, Shilong Chen, Yongcui Wang

**Affiliations:** 1Key Laboratory of Plateau Biological Adaptation and Evolution, Northwest Institute of Plateau Biology, Chinese Academy of Sciences, Xining 810008, China; yuanrui@nwipb.cas.cn (R.Y.); slchen@nwipb.cas.cn (S.C.); 2University of Chinese Academy of Sciences, Beijing 100049, China; 3Institute of Sanjiangyuan National Park, Chinese Academy of Sciences, Xining 810008, China; 4Qinghai Provincial Key Laboratory of Crop Molecular Breeding, Northwest Institute of Plateau Biology, Chinese Academy of Sciences, Xining 810008, China

**Keywords:** DNA methylation, group lasso, CTRP and CCLE, lung cancer, EGFR inhibitors effectiveness related methylation sites

## Abstract

The emergence of drug resistance is one of the main obstacles to the treatment of lung cancer patients with EGFR inhibitors. Here, to further understand the mechanism of EGFR inhibitors in lung cancer and offer novel therapeutic targets for anti-EGFR-inhibitor resistance via the deep mining of pharmacogenomics data, we associated DNA methylation with drug sensitivities for uncovering the methylation sites related to EGFR inhibitor sensitivity genes. Specifically, we first introduced a grouped regularized regression model (Group Least Absolute Shrinkage and Selection Operator, group lasso) to detect the genes that were closely related to EGFR inhibitor effectiveness. Then, we applied the classical regression model (lasso) to identify the methylation sites associated with the above drug sensitivity genes. The new model was validated on the well-known cancer genomics resource: CTRP. GeneHancer and Encyclopedia of DNA Elements (ENCODE) database searches indicated that the predicted methylation sites related to EGFR inhibitor sensitivity genes were related to regulatory elements. Moreover, the correlation analysis on sensitivity genes and predicted methylation sites suggested that the methylation sites located in the promoter region were more correlated with the expression of EGFR inhibitor sensitivity genes than those located in the enhancer region and the TFBS. Meanwhile, we performed differential expression analysis of genes and predicted methylation sites and found that changes in the methylation level of some sites may affect the expression of the corresponding EGFR inhibitor-responsive genes. Therefore, we supposed that the effectiveness of EGFR inhibitors in lung cancer may be improved by methylation modification in their sensitivity genes.

## 1. Introduction

Epidermal Growth Factor Receptor (EGFR) is a tyrosine kinase receptor which plays a crucial role in many carcinogenic processes, including tumor cell proliferation, angiogenesis, invasion and metastasis, inhibition of apoptosis, etc. [1,2,3]. Currently, numerous studies have shown that EGFR is overexpressed or abnormally expressed in many different types of tumors [4,5,6,7], including in more than 80% of non-small cell lung cancers (NSCLC). Therefore, EGFR has become an attractive therapeutic target for NSCLC patients, but drug resistance invariably emerges. Despite initially responding to EGFR tyrosine kinase inhibitors, most patients will develop resistance to the drug within 1 to 2 years after the first treatment [8,9,10]. In previous work, researchers have made significant contributions to exploring the mechanism of EGFR inhibitor resistance. Balak et al. [11] examined tumor cells from 16 patients with acquired drug-resistant lung cancer by molecular analysis; in about 50% of these cases, resistance was due to the occurrence of a secondary mutation in EGFR (T790M). Engelman et al. [12] constructed 18 lung cancer cell lines resistant to Gefitinib or Erlotinib (EGFR tyrosine kinase inhibitors) and found that MET amplification might promote cellular resistance to EGFR target therapies, and Guix et al. [13] found that the absence of expression of Insulin-Like Growth Factor (IGF) is associated with the resistance to EGFR inhibitors in A431 lung squamous cancer cells. Meanwhile, with the help of high-throughput sequencing technology, researchers attempted to predict drug response based on statistical theory by using large-scale pharmacogenomics data. For example, the EGFR mutation profile of cell lines was used by Guan et al. [14] to investigate the relationship between gene mutations and response to Lapatinib (EGFR inhibitor), and found that cell lines with EGFR mutations were more sensitive to Lapatinib on both predicted and observed data; Chiu et al. [15] designed a deep learning model and predicted that Afatinib and Gefitinib, two inhibitors of EGFR, performed better in NSCLC with EGFR mutation than without EGFR mutation by using a large number of tumor samples from the Cancer Genome Atlas (TCGA) database. Wei et al. [16] used the gene expression and drug response data of 144 lung cancer cell lines to prioritize 549 drugs whose response was correlated with up-regulated genes in Gefitinib-resistant cell lines, and the top drugs were evaluated for their response in these cell lines. Therefore, because most patients will eventually develop resistance to EGFR inhibitors, it is an important challenge to explore the resistance mechanism of EGFR inhibitors by deep mining the pharmacogenomics data to identify new therapeutic targets.

In recent years, as research deepens, DNA methylation has become one of the most studied oncogene regulatory mechanisms. DNA methylation, as a common epigenetic modification, occurs on the cytosine bases of the CpG sequence under the premise of unchanged DNA sequences, which will affect the transcription of downstream genes [17,18,19,20]. Many studies have now provided evidence that specific methylation changes can affect the response to different cancer treatments; thus, DNA methylation could be a valuable resource to study drug effectiveness in cancers. One prominent example is that in human T-cell leukemia cell lines, researchers have found a strong correlation between demethylation of the 5′ region of MDR1 gene and multi-drug resistance. MDR1 is a multi-drug resistance gene encoding P-glycoprotein, which is involved in transporting substances across cellular membranes. As we all know, P-glycoprotein is often overexpressed in cancer cells, which is related to the increased efflux of cytotoxic drugs [21]. Later, Worm et al. [22] discovered that in MDA-MB-231 breast cancer cell lines, methylation of the CpG island in the promoter region of SLC19A1 can lead to methotrexate resistance; meanwhile, it has been found that in xenografts from ovarian and colon cancer cell lines, treatment with a demethylating agent induces the re-expression of MLH1, thereby sensitizing the xenografts to cisplatin, carboplatin, temozolomide, and epirubicin [23]. These studies suggested that some genes related to drug response regulation are methylated and silenced, thus DNA methylation can be used as a useful resource to study the mechanism of drug resistance, and offer a novel way to identify the targets for anti-drug resistance.

In 2012, multiple research groups from the Broad Research Institute, Dana-Farber Cancer Research Institute, and Novartis Biomedical Research Institute jointly completed the Cancer Cell Line Encyclopedia (CCLE) project, which carried out a large-scale deep sequencing of 947 human cancer cell lines covering more than 30 tissue sources, and integrated genetic information, such as DNA mutations, gene expression, and chromosome copy number [24]. In 2019, the CCLE database received a major update, including newly released DNA methylation data, whole genome sequencing data, and RNA-seq data [25]. The Cancer Therapeutics Response Portal (CTRP) is a public and interactive portal that covers cancer cell line compound sensitivity and genetic or lineage characteristics. Furthermore, CTRP makes full use of CCLE’s common data, making the two projects excellently complement each other [26]. These valuable databases provide a great opportunity to study the relationship between DNA methylation and drug response. With a deep understanding of the regulatory role of DNA methylation in drug responses in cancer, the novel therapeutic targets could be unveiled for the inhibition of drug resistance.

Inspired by the above observations, here, we proposed a novel computational framework to predict the methylation sites related to EGFR inhibitor sensitivity genes. The purpose was to explore possible mechanisms that affected the EGFR inhibitor response in cancers and provide novel targets for anti-EGFR inhibitor resistance. Specifically, we first identified genes that were closely related to drug effectiveness through a Group Least Absolute Shrinkage and Selection Operator (group lasso) regression model. It is worth mentioning that for the special case of linear regression, the lasso solution is not satisfactory when there are not only continuous variables, but also classified prediction variables (factors), because it only selects a single dummy variable instead of the entire factor. Therefore, group lasso overcomes this issue by introducing a suitable extension of the lasso penalty [27,28,29,30]. Particularly, based on the group lasso algorithm, it is reasonable to implement penalties on model parameters by gene grouping. Then, we predicted the methylation sites associated with the effectiveness of EGFR inhibitor genes by the lasso regression model, which performs both variable selection and regularization to improve the prediction accuracy and enhance the interpretability of the statistical model [31,32,33,34]. Furthermore, to detect the function of the predicted methylation sites related to the effectiveness of EGFR inhibitors, GeneHancer [35] and Encyclopedia of DNA Elements (ENCODE) databases were introduced [36]. The genome position analysis indicated that the predicted methylation sites related to EGFR inhibitor sensitivity genes share genome loci with regulatory elements, including promoter, enhancer, and translation factor binding regions (TFBS). Finally, we performed differential expression analysis on EGFR inhibitor sensitivity genes and the predicted methylation sites, and found some examples that displayed the changes in methylation level of predicted methylation sites, which may lead to changes in the expression level of corresponding responsive genes, and then affect the effectiveness of drugs in lung cancer. This result suggested a possible way to repress EGFR inhibitor resistance, that is, the effectiveness of EGFR inhibitors in lung cancer might be improved by methylation modification of genes that are closely related to EGFR inhibitors resistance.

## 2. Materials and Methods

### 2.1. Materials

The benchmark dataset used for the validation of the models came from the well-known cancer genomics resource, CTRP. The CTRP database deposited the AUCDR (area under the dose-response curve) of 481 drugs across 664 cancer cell lines, which is an important indicator of how a particular cancer cell line responds to a given anti-cancer drug [37]. The download link for this data is: http://portals.broadinstitute.org/ctrp.v2.1/ (accessed on 31 May 2018). In this paper, we only focus on the sensitivity of EGFR inhibitors in lung cancer cell lines. Therefore, we extracted the AUCDR values of 10 EGFR inhibitors (WZ8040, WZ4002, Vandetanib, PD_153035, Neratinib, Lapatinib, Gefitinib, Erlotinib, Canertinib, Afatinib) in 121 lung cancer cell lines. At the same time, we used two cancer omics data, namely “CCLE_RNAseq_rsem_genes_tpm_20180929.txt” and “CCLE_RRBS_cgi_CpG_clusters_20181119.txt”. The download links for them are: https://data.broadinstitute.org/ccle/CCLE_RNAseq_rsem_genes_tpm_20180929.txt.gz (accessed on 31 May 2018), and https://data.broadinstitute.org/ccle/CCLE_RRBS_cgi_CpG_clusters_20181119.txt.gz (accessed on 31 May 2018), respectively. The RNA-seq data includes 57,820 gene expressions across 1019 cancer cell lines, and the DNA methylation data contains beta values of 1,208,342 methylation sites across 843 cancer cell lines. Here, the gene annotation file from the Ensemble database was introduced, which was to applied to screen the methylation sites located in non-coding regions. The download link for the gene annotation file is: http://ftp.ensembl.org/pub/release-83/gtf/homo_sapiens/Homo_sapiens.GRCh38.83.gtf.gz (accessed on 31 May 2018). Finally, the expression values of 24,643 genes and the beta values of 418,677 methylation sites across 153 common lung cancer cell lines were retained for further analysis. That is, 418,677 methylation sites in the non-coding regions were trained in the lasso model.

### 2.2. Methods

#### 2.2.1. Identification of Drug Responsive Genes via Group Lasso Regularization

It has been reported that genes are not independent of each other, co-expressed genes may have similar biological functions, and the effect of grouping genes is relatively strong [38]. Here, to overcome the gene group effect, the group lasso regression model [27], which aims to identify the drug responsive genes based on drug screening experimental results and RNA-seq data, was introduced (Figure 1A,B). The formula of the group lasso is as follows:$$min_{\beta \;}\frac{1}{2}\|y_{i}-\overset{m}{\underset{l=1}{\sum }}x_{i}^{\left(l\right)}\beta^{\left(l\right)}{\|}_{2}^{2}+\lambda \overset{m}{\underset{l=1}{\sum }}\sqrt{\rho l}{\left\|\beta^{\left(l\right)}\right\|}_{2}$$
where $y_{i}$ is the AUCDR value of *i*-th EGFR inhibitors across 153 cancer cell lines, and $x_{i}^{\left(l\right)}$ is the *i*-th gene expression in group l with rows as the cancer cell lines. $\beta^{\left(l\right)}$ is the coefficient vector of that group and ρl is the length of $\beta^{\left(l\right)}$. Meanwhile, $\sqrt{\rho l}$ is the weight of each group and λ is the regularization parameter. In this paper, the group lasso model was implemented via an “SGL” R package (main parameters: type = “linear”, alpha = 0.9). The groups of genes were obtained by the R “hclust” function with “ward.D2” as the hierarchical clustering method. To understand which biological functions and important pathways the predicted genes were enriched in, we performed a Gene ontology (GO) and Kyoto Encyclopedia of Genes and Genomes (KEGG) enrichment analysis via DAVID Bioinformatics Resources [39,40].

#### 2.2.2. Prediction of DNA Methylation Sites Related to Drug Responsive Genes via Lasso Regression

Once we obtained genes related to the sensitivity of each EGFR inhibitor (Figure 1C), DNA methylation sites related to drug responsive genes can be obtained. Specifically, the lasso regression model was introduced to predict the methylation sites related to the drug-responsive genes (Figure 1D). Lasso was first proposed by Robert Tibshirani in 1996. It is a linear regression method that adopts L1 regularization, which makes partial learned feature weights equal to 0 so as to achieve the purpose of sparsity and feature selection [32,33]. Here, the lasso model was implemented via a “glmnet” R package, and the best lambda was determined by a grid search. The input and output of the lasso regression model were the beta value of the methylation site and the expressions of genes associated with a given EGFR inhibitor sensitivity across the common 153 lung cancer cell lines. The lasso model was implemented on each given gene related to EGFR inhibitor sensitivity. To show the biological usefulness of predicted methylation sites, we checked whether they were located in some important regulatory elements, including enhancers, promoters, or TFBS, through a database search. Specifically, we first used the GeneHancer database, a novel database of human enhancers and their inferred target genes, to see whether the methylation site falls in the enhancer region [35]. Then, the ENCODE database, which provides a wealth of data and clarifies the role of functional elements in the human genome [36], was applied to check whether the identified methylation sites were located in the promoter region or the TF binding region. Subsequently, the Pearson Correlation Coefficient (PCC) was calculated between the beta value of the predicted methylation site and the drug responsive genes.

#### 2.2.3. Differential Expression Analysis

To detect the regulatory role of the predicted methylation sites, we performed differential expression analysis on 24,643 genes and their associated methylation sites in sensitive and resistant cancer cell lines. Here, we classified the cancer cell lines as sensitive or resistant according to the AUCDR data, and Table 1 shows the thresholds of classification and the number of cancer cell lines in each group for 10 EGFR inhibitors, respectively. Specifically, we first performed differential expression analysis on 24,643 genes to find the differentially expressed genes for 10 EGFR inhibitors, respectively. Subsequently, according to the prediction results of the lasso regression model, the methylation sites closely related to these differentially expressed genes were obtained. Finally, differential expression analysis was performed on these methylation sites in the same sample, and the methylation sites with significantly different methylation levels were selected as the candidates displaying the possible regulatory relationships with drug responsive genes. The differential expression analysis was implemented via a “limma” R package.

## 3. Results

### 3.1. The Drug Responsive Genes Are Enriched in Lung Cancer Developmental Processes

In order not to ignore the influence of gene co-expression, the 24,643 genes were firstly divided into 20 groups through hierarchical clustering analysis. Figure 2A shows the results of gene clustering, and different groups are indicated by different colors. Figure 2B shows the number of genes contained in each group, and we can see that the gene group effect is strong.

The group lasso model (equation X) was applied to the grouped RNA-seq data and the sensitivity data (AUCPR) of EGFR inhibitors and the grouped drug responsive genes were obtained. The GO and KEGG enrichment analyses on these predicted responsive genes were performed, and the functions and pathways with a *p* value of less than 0.05 are shown in Figure 3. Here, we only focus on biological processes related to EGFR inhibitor sensitivity (Figure 3A). Figure 3A shows that, among the eight biological processes related to EGFR inhibitor sensitivity, the genes are well enriched in response to drug and lung development. Figure 3B shows the enrichment of six KEGG pathways. We found that the sensitivity genes of 10 EGFR inhibitors were significantly correlated with cancer pathways.

### 3.2. The Predicted Methylation Sites Related to EGFR Inhibitors Sensitivity Share Locus with Regulatory Elements

The methylation sites related to EGFR inhibitor sensitive genes were obtained by lasso regression of expressions of drug responsive genes. Through GeneHancer and ENCODE databases, we checked whether those methylation sites share loci with regulatory regions (Figure 4). Figure 4A shows the proportion of predicted methylation sites located in the enhancer, promoter, or TF binding regions of corresponding responsive genes. We can see that about 6% of predicted methylated sites are located in the enhancer or promoter region, and the variations are small for all ten EGFR inhibitors. In contrast, the variation of predicted methylation sites falling into the TF binding region is higher. For example, about 44% of predicted methylation sites related to the sensitivity to the drug Lapatinib fall in the TF binding region; however, only about 4% of predicted methylation sites related to WZ8040 share loci with the TF binding region. Figure 4B shows the number of responsive genes with relevant methylation sites located in a regulatory element. Detailed information about this part is presented in the Appendix A. Subsequently, we calculated the PCC between the beta value of predicted methylation sites located in the regulatory region and the expression of responsive genes. For comparison, we randomly selected methylation sites and corresponding genes from the lasso prediction results as the control group (Figure 5A). As a result, compared with the control group, the absolute values of PCC in regulatory groups are higher. Figure 5B shows the number of predicted methylation sites located in the regulatory region with absolute spearman PCC values greater than 0.3. Our results suggested that the well-correlated predicted methylation sites located in promoters are more than those in the enhancer and TF binding region, suggesting the important role of methylation sites in the promoter region of responsive genes. In order to further explore the role of methylation sites in the resistance of EGFR inhibitors, we performed differential expression analysis on responsive genes and predicted methylation sites, respectively. Figure 6A shows the fold change of gene expression value and −log10 *p* value analysis on genes and methylation sites for ten EGFR inhibitors (Appendix A). The number of genes with significant differences in expression with an absolute value of logFC > 0.5 and a *p* value < 0.05 are shown in Figure 6B. Taking Erlotinib as an example, the number of differentially expressed genes is up to 104, but only 28 differentially expressed genes are found to be correlated with the effectiveness of WZ4002. Based on the prediction results of the lasso model, methylation sites associated with these differentially expressed genes can be found. Through further differential expression analysis on these methylation sites, we identified several pairs of genes and methylation sites that may be involved in regulation (Appendix A). For example, we found a significant difference in the expression value of MARVELD2 among Lapatinib-sensitive (DV90_LUNG) and Lapatinib-resistant (SBC5_LUNG) lung cancer samples. Through the lasso regression model, methylation site “5:68711681” is closely related to the MARVELD2 gene. Furthermore, the beta value of methylation site “5:68711681” shows a significant difference (absolute value of logFC > 0.4 and *p* value < 0.05) among Lapatinib-sensitive and Lapatinib-resistant lung cancer samples. It indicated that the low methylation of “5:68711681” may promote the expression of the Lapatinib-related gene MARVELD2 (Figure 7A); on the contrary, the high methylation of this site may inhibit MARVELD2 gene expression, and then affect the effectiveness of Lapatinib in lung cancer cells (Figure 7B). Therefore, one of the important factors affecting the expression of drug responsive genes may be the level of the methylation sites in their promoter region.

## 4. Discussion

Systematic study of the relationship between cancer cells and anticancer therapies could provide novel therapeutic targets for early clinical trials. In this paper, our main contribution is to explore the possible resistance mechanism of EGFR inhibitors from the perspective of DNA methylation. Therefore, we took a series of measures to find some new potential relationships. Firstly, to overcome the group effect of genes, a group lasso regression model was introduced to detect the genes closely related to drug responses in cancers. We performed GO and KEGG enrichment analyses on the sensitivity genes for 10 EGFR inhibitors, respectively. The results showed that many sensitivity genes were significantly enriched in the “response to drug”. For example, we found that the MET gene was significantly associated with the effectiveness of Erlotinib. Engelman et al. [12] constructed lung cancer cell lines resistant to Erlotinib and found that MET amplification was detected in about 22% of the samples, suggesting that MET amplification may be one of the important factors leading to Erlotinib resistance in lung cancer cell lines. Similarly, Jakobsen et al. [41] established Erlotinib-resistant lung adenocarcinoma cell lines and produced 14 resistant subclones, and found that approximately 42% of the subclones exhibited MET amplification. As we all know, EGFR-mutated non-small-cell lung cancer was highly sensitive to a variety of EGFR inhibitors [42,43,44], and our results also predict that EGFR was the responsive gene of Vandetanib, Lapatinib, Gefitinib, Canertinib, and Afatinib. López-Ayllón et al. [45] identified that the expressions of UGT1A6, MET, and LCN2 genes were correlated with the effectiveness of Erlotinib, and especially with the low MET expression level showing the strongest correlation. In this study, these genes were predicted to be associated with Erlotinib sensitivity in non-small-cell lung cancer. Second, the methylation sites closely linked to these responsive genes were predicted by the lasso regression model, which is one of the classical regression algorithms. Third, the predicted methylation sites were related to regulatory elements and the correlation analysis was performed on them. The results showed that the predicted methylation sites associated with EGFR inhibitor-responsive genes seemed to be more related to the promoter region of their associated gene. Over the past decade, it has become increasingly clear that aberrant promoter methylation seems to be related to the loss of gene function. For example, hypermethylation of the promoter region is a major mechanism to suppress tumor genes and silence other cancer-related genes in many human cancers. Baldwin et al. discovered that promoter hypermethylation may be an important way to inactivate the BRCA1 tumor suppressor gene, and there was a clear association between the BRCA1 gene mutation and hereditary ovarian cancer [46]. Martínez-Galán et al. [47] reported that the hypermethylation of the promoter region of the ESR1 gene in breast cancer patients will affect the expression of the estrogen receptor protein, and concluded that epigenetic markers in plasma might be a new target for anticancer therapy, especially in endocrine therapy. Mijnes et al. [48] identified that the promoter region of the RBBP8 gene was almost hypermethylated in bladder cancer. However, RBBP8 has been shown to play a role in the repair of DNA double strand breaks mediated by homologous recombination, which is known to make cancer cells sensitive to PARP1 inhibitors. According to previous studies, the methylation of CpG sites in the gene promoter region affects the activity of gene transcription in three ways: DNA sequence methylation directly hinders the binding of transcription factors; the CpG-binding protein binds to methylated CpG site and interacts with other transcription inhibitors; condensation of chromatin structure blocks the binding of transcription factors to their regulatory sequences [49,50,51]. Therefore, inspired by the above observations, it is reasonable to hypothesize that methylation in some important regulatory regions of EGFR inhibitor-responsive genes may lead to changes in the effectiveness of the drug. In this paper, to further validate that, we performed differential expression analysis on responsive genes and their associated methylation sites in EGFR inhibitor-sensitive and -resistant lung cancer cell lines. As a result, we found some examples that displayed changes in the methylation level of predicted methylation sites which may lead to changes in the expression level of corresponding responsive genes, and then affect the effectiveness of drugs in lung cancers. For example, the hypermethylation of site 5:68711681 was linked with the differential expression of MARVELD2 in Lapatinib-sensitive and -resistant lung cancer cell lines. MARVELD2 encodes the trillulin protein, which is a membrane protein found in tight junctions between epithelial cells [52]. Previous studies have confirmed that tight junctions are an important part of signal transduction and the cellular barrier, which refers to the barrier separating the cell membrane. Indeed, tight junctions are closely associated with various types of cancers and drug delivery. Since the drug must pass through the epithelium and the inner membrane to reach the target tissue, that is, the ability of the drug to pass through these membranes is directly related to the effectiveness of the drug [53,54,55]. Therefore, we conducted a hypothesis that the level of methylation may affect the expression of EGFR inhibitors of responsive genes, and then influence the effectiveness of drugs in cancer.

Due to the continuous change and development of high-throughput sequencing technology, researchers have made more attempts to use biological knowledge to understand the regulatory mechanisms of gene expression from the perspectives of genomics, proteomics, metabolomics, and other omics, and to explore the internal rules of human disease diagnosis and drug therapy, so as to elucidate the genetic characteristics of life. Here, we associated DNA methylation data with drug sensitivity data to detect the possible regulatory relationship between methylation sites and EGFR inhibitor-responsive genes. In summary, this study demonstrated the correlation between DNA methylation and the effectiveness of EGFR inhibitors, and suggested that DNA methylation might be one of the important regulatory factors affecting the sensitivity of EGFR inhibitors in lung cancer patients.

It is worth mentioning that this study has some limitations. On the one hand, the lung cancer samples extracted in this study are derived from different tissues, which could have a certain impact on our results (small_cell_carcinoma: 49; adenocarcinoma: 39; non_small_cell_carcinoma: 22; squamous_cell_carcinoma: 16; large_cell_carcinoma: 14; mixed_adenosquamous_carcinoma: 4; NS: 3; undifferentiated_carcinoma: 1; mucoepidermoid_carcinoma: 1). Our research results are only based on the current data. In the future, we will try to test our model on lung cancer cell lines extracted from other databases, such as GDSC, in order to obtain more robust conclusions. On the other hand, there are few EGFR inhibitor-sensitive genes that have been experimentally verified. Therefore, we look forward to verifying our conclusion with more experimental data in the future.

## 5. Conclusions

In the study, we introduced the CTRP and CCLE as the basic databases. First, we constructed the group lasso regression models of 10 EGFR inhibitors, respectively, to obtain the gene sets related to drug sensitivity. Then, the methylation sites closely related to these genes were predicted based on the lasso regression algorithm. Finally, we further combined the location information of predicted methylation sites and the correlation analysis with drug-related genes. The results show that, compared with enhancers and transcription factors, EGFR inhibitor-related genes have a stronger correlation with the methylation sites in the promoter region of this gene. Therefore, the effect of EGFR inhibitors in lung cancer may be affected by regulating the methylation level of the drug-sensitive gene promoter region.

## Figures and Tables

**Figure 1 biomolecules-11-01042-f001:**
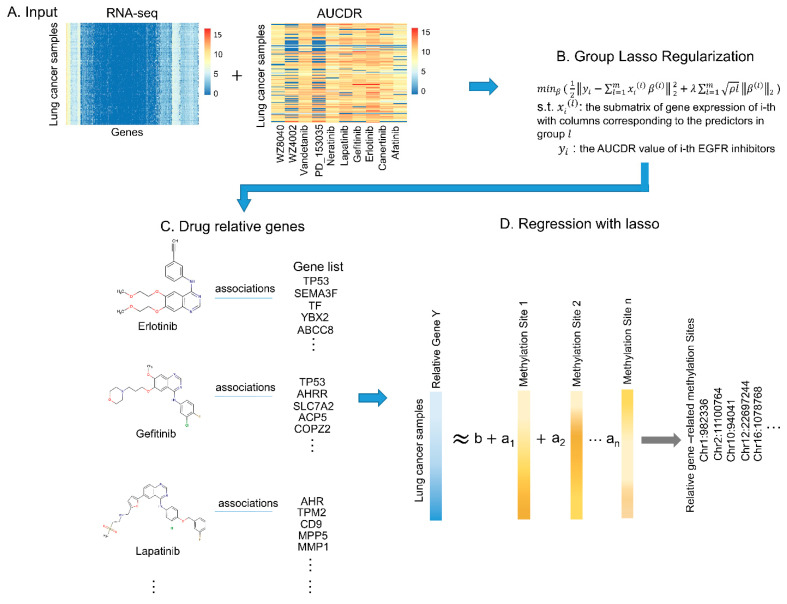
Machine learning flowchart. (**A**) Pharmacogenomic data including lung cancer mRNA expression and EGFR inhibitor drug sensitive data were introduced. These data were from CCLE and CTRP databases, respectively. (**B**) The prediction models of EGFR inhibitor-response genes were constructed based on group lasso algorithm. (**C**) The gene sets associated with each of the 10 EGFR inhibitor responses were predicted. (**D**) Using lasso model to predict methylation sites related with EGFR inhibitor sensitivity.

**Figure 2 biomolecules-11-01042-f002:**
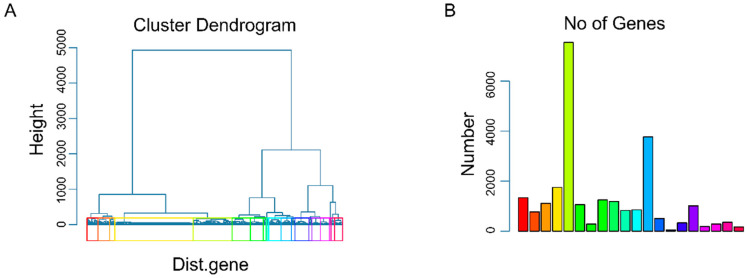
The result of gene clustering. (**A**) Divide genes into 20 groups as an important parameter for the construction of group lasso. (**B**) The number of genes in each group shown as a histogram.

**Figure 3 biomolecules-11-01042-f003:**
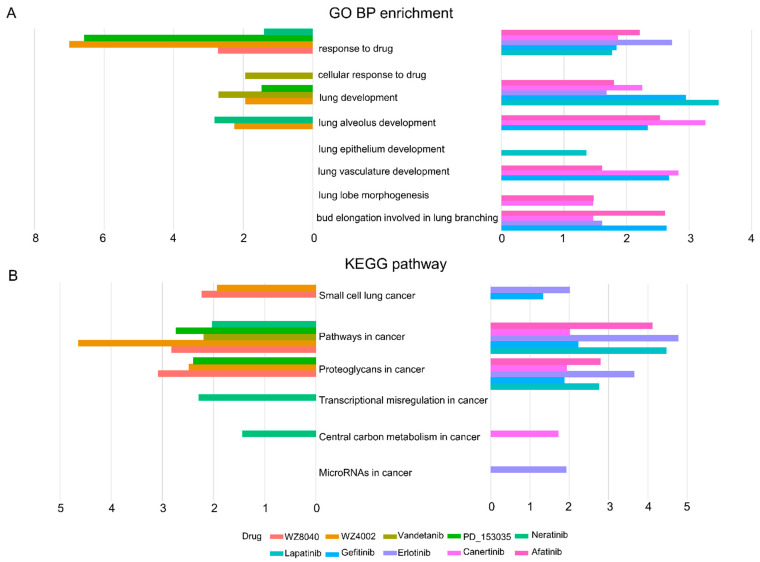
Biological process and pathway enrichment of EGFR inhibitor-responsive genes. (**A**) Introduce the GO database to view the predicted EGFR inhibitor-responsive gene enrichment and view which were related to the biological processes related to drugs and lungs. (**B**) Introduce the KEGG database to check which gene pathways related to cancer were related to the predicted EGFR inhibitor-responsive genes.

**Figure 4 biomolecules-11-01042-f004:**
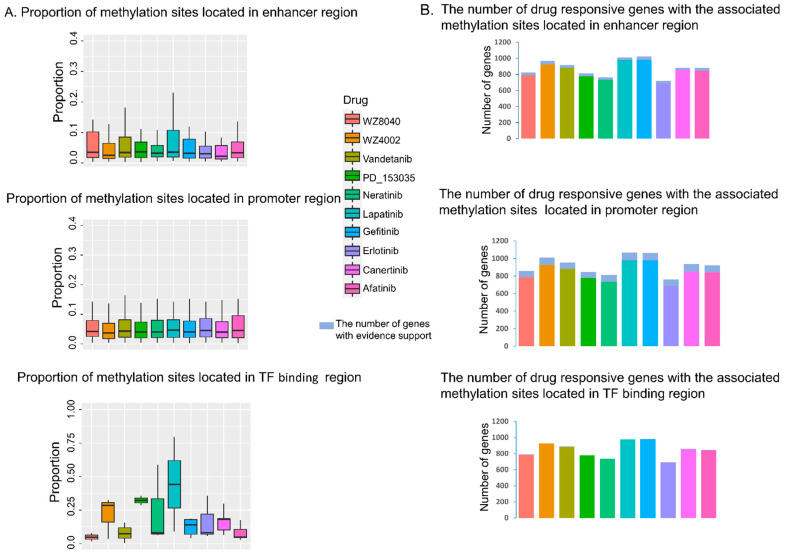
Location information of methylation sites related to EGFR inhibitor response. (**A**) The proportion of predicted methylation sites located in enhancer, promoter, or TF binding regions. (**B**) The light blue bar represents the number of responsive genes supported by evidence; that is, the methylation sites closely associated with the responsive genes were located in important regulatory regions (enhancer, promoter, transcription factor binding) for each of the 10 EGFR inhibitors, respectively.

**Figure 5 biomolecules-11-01042-f005:**
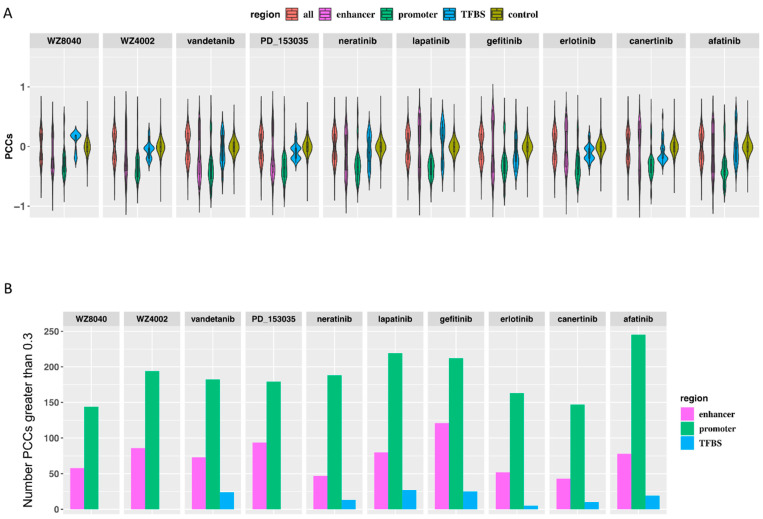
Spearman correlation coefficients between the predicted EGFR inhibitor-responsive genes and the associated methylation sites. (**A**) According to the location information of methylation sites, they were divided into five groups, namely, the enhancer group, the promoter group, and the TF binding group, which were located in the three regulatory groups and the control group. (**B**) The barplot shows the number of PCCs related to enhancer, promoter, and TF binding regions greater than 0.3.

**Figure 6 biomolecules-11-01042-f006:**
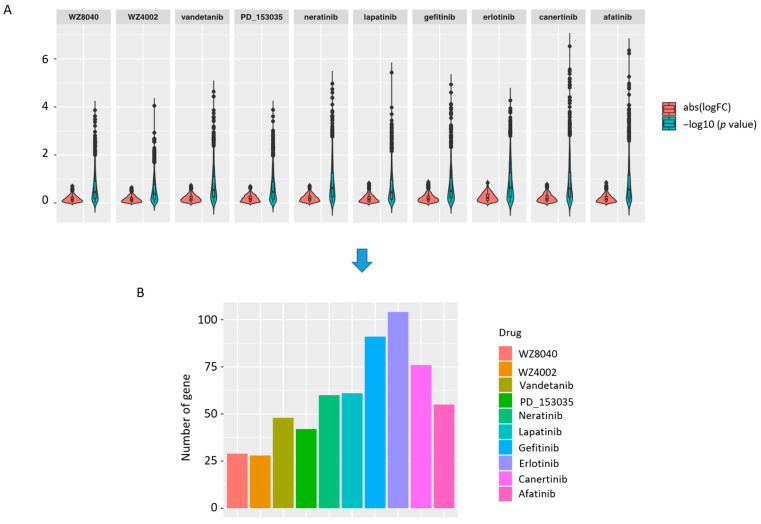
Results of difference analysis. (**A**) The violin plot shows the absolute value of logFC and −log10(*p* value) for each EGFR inhibitor. (**B**) The number of absolute value of logFC > 0.5 and the *p* value < 0.05 for each EGFR inhibitor.

**Figure 7 biomolecules-11-01042-f007:**
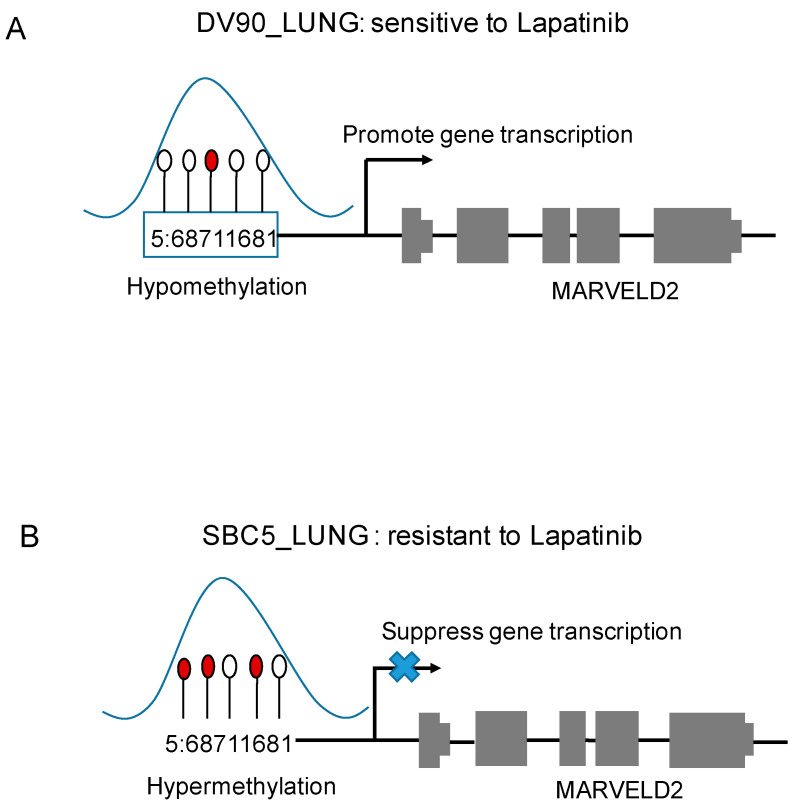
The relationship between methylation and gene expression. (**A**) In lung cancer samples sensitive to Lapatinib (DV90_LUNG), methylation site—5:68711681 hypomethylation may promote the transcription of MARVELD2 gene. (**B**) In lung cancer samples resistant to Lapatinib (SBC5_LUNG), methylation site—5:68711681 hypermethylation may suppress the transcription of MARVELD2 gene.

**Table 1 biomolecules-11-01042-t001:** Classification conditions of EGFR inhibitor samples.

Drug	Sensitivity	Resistance	No. of Sensitive Cell Lines	No. of Resistance Cell Lines
WZ8040	>0 & <10.5	>12	22	27
WZ4002	>0 & <12	>13	18	28
Vandetanib	>0 & <11.5	>13.5	25	26
PD-153035	>0 & <12.5	>14.5	19	21
Neratinib	>0 & <10	>13	24	33
Lapatinib	>0 & <11.5	>14	21	24
Gefitinib	>0 & <11	>13.5	19	21
Erlotinib	>0 & <11.5	>14.5	19	21
Canertinib	>0 & <10	>13	21	33
Afatinib	>0 & <9	>11.5	27	30

## Data Availability

Not applicable.

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
