# Peer review of "Computational Probing the Methylation Sites Related to EGFR Inhibitor-Responsive Genes"

_biomolecules, 2021, doi:10.3390/biom11071042_

Round 1

Reviewer 1 Report

This manuscript presents a novel computational framework to predict the methylation sites related to EGFR inhibitor sensitivity genes. However, there are several concerns and questions that would improve the paper.

  1. Figure 1D, how to choose the candidate methylation sites included in lasso model? In section 2.2.3, the author used the 24,643 drug responsive genes, which means, the authors included all methylation sites overlapped with promoter regions/enhancer regions of those drug responsive genes? Otherwise, how can you filter them?
  2. The one of critical issue in hierarchical clustering method is to find the cut point to find the ideal number of clusters. How did you choose 20 clusters? It seems that there are lots of clusters having small number of genes. Could you please clarify the criteria? How many genes did you use to cluster? I guess that you may need to consider only significantly meaningful genes for cluster instead of using all genes.
  3. Figure 5B, could you make the bar-plot with the number of PCCs related to enhancer, promoter, and TF binding region greater than 0.6 or larger value? To me, the PCC threshold 0.3 seems to low.

Reviewer 2 Report

The authors investigated the association between DNA methylation with drug sensitivities
for uncovering the methylation sites related with EGFR inhibitors sensitivity genes.

Specific points to be addressed:

  • Page 3 http://portals.broadinstitute.org/ctrp/?page=%20-%20ctd2Cluster give me a empty page. Could the authors give us the correct link?
  • The same for https://data.broadinstitute.org/ccle/CCLE_RRBS_cgi_CpG_clusters_20181119. txt.gz
  • -page 7 “The methylation sites………was obtained…….” should be “The methylation sites………were obtained…….”
  • Figure 4. The authors should indicate the labels for x-axis
  • Figure 4b is not clear. What do the light blue bars represent above the main bars?
  • I think the authors should be discuss more the sensitivity genes
  • The authors should include a section discussing the limit of their study. For example, cell lines derive by different tissues, according to the authors this aspect could influence the results?
  • The authors should include in supplementary materials the code of regression lasso model

Round 2

Reviewer 1 Report

The authors addressed all the issues I have addressed. 

Good luck! 

Reviewer 2 Report

The manuscript has been significantly improved